# Appearance of a Solitary Wave Particle Concentration in Nanofluids under a Light Field

**DOI:** 10.3390/nano11051291

**Published:** 2021-05-14

**Authors:** Abram I. Livashvili, Victor V. Krishtop, Polina V. Vinogradova, Yuriy M. Karpets, Vyacheslav G. Efremenko, Alexander V. Syuy, Evgenii N. Kuzmichev, Pavel V. Igumnov

**Affiliations:** 1Institute of Natural Sciences, Far Eastern State Transport University, 47, Seryshev St., 680021 Khabarovsk, Russia; livbru@mail.ru (A.I.L.); vpolina17@hotmail.com (P.V.V.); kjum1947@mail.ru (Y.M.K.); oblako3@yandex.ru (V.G.E.); 2Department of General Physics, Perm National Research Polytechnic University, 29, Komsomolsky Prospekt, 614990 Perm, Russia; 3Department of General Physics, Moscow Institute of Physics and Technology, 9, Institutskiy Per., 141701 Dolgoprudny, Russia; alsyuy271@gmail.com; 4Institute of Materials Technology of Khabarovsk Centre of FEC the Russian Academy of Sciences, 153, Tihookeanskaya St., 680042 Khabarovsk, Russia; e_kuzmichev@mail.ru (E.N.K.); 407320@mail.ru (P.V.I.)

**Keywords:** nanofluid, nonlinear dynamics, colloidal suspension, solitary wave, Burgers–Huxley equation

## Abstract

In this study, the nonlinear dynamics of nanoparticle concentration in a colloidal suspension (nanofluid) were theoretically studied under the action of a light field with constant intensity by considering concentration convection. The heat and nanoparticle transfer processes that occur in this case are associated with the phenomenon of thermal diffusion, which is considered to be positive in our work. Two exact analytical solutions of a nonlinear Burgers-Huxley-type equation were derived and investigated, one of which was presented in the form of a solitary concentration wave. These solutions were derived considering the dependence of the coefficients of thermal conductivity, viscosity, and absorption of radiation on the nanoparticle concentration in the nanofluid. Furthermore, an expression was obtained for the solitary wave velocity, which depends on the absorption coefficient and intensity of the light wave. Numerical estimates of the concentration wave velocity for a specific nanofluid—water/silver—are given. The results of this study can be useful in the creation of next-generation solar collectors.

## 1. Introduction

In recent years, the optical properties of colloidal suspensions (nanofluids) have been actively studied [1,2,3,4,5,6]. Researchers are particularly interested in nonlinear optical effects that are realized in such media. In particular, studies have focused on four-wave interactions and the self-action of light waves [7,8,9,10]. Without detailed knowledge of the optical properties of nanofluids, it is impossible to create next-generation solar collectors [11,12,13,14,15]. For example, [16] summarized the results of studies on the nanocolloids of ionic liquids (i.e., ionic liquids with nanoparticles in suspension), which can be directly applied to convective heat transfer. In [17], machine learning was used to develop Gaussian process regression models to describe the statistical correlations between the thermal conductivity and physical parameters of two-phase nanofluid components. For this purpose, approximately 300 samples of nanofluids, dispersions of metal, and ceramic nanoparticles in water, ethylenecol, and transformer oil have been investigated. The modeling approach demonstrates a high degree of accuracy and stability, facilitating efficient and inexpensive thermal conductivity estimates. Work [18] considered a liquid consisting of a stable colloidal suspension of magnetic maghemite nanoparticles in water. It has been found that these nanoparticles constitute an excellent absorber of solar radiation and simultaneously an amplifier of thermoelectric power output with a very small volume fraction when the liquid is heated from above. These results demonstrate that the investigated nanofluid has great potential as a coolant for the co-production of heat and energy in completely new hybrid flat solar thermal collectors, for which top heating geometry is required. The main mechanisms for optical nonlinearity in these cases are the phenomena of thermal diffusion and electrostriction of nanoparticles [19,20]. Despite the many studies on this problem [21,22,23,24,25], several questions still remain. In particular, the dynamics of the concentration of nanofluid particles are unknown in the presence of concentration dependences on the coefficients of the thermal conductivity, viscosity, and absorption of radiation of the medium. Providing a theoretical description of the processes of heat and mass transfer for the nanofluid and radiation system is fraught with serious mathematical difficulties that are associated with the search for analytical solutions of the corresponding nonlinear equations. In this study, we developed a theoretical model for the dynamics of the concentration of nanoparticles in a liquid-phase medium when subjected to constant-intensity laser irradiation. Further, the study considers the dependence of the coefficients of absorption of radiation, thermal conductivity, and viscosity of the medium on the concentration of nanoparticles. It should be noted that, in the works cited above, the dynamics of the concentration of nanoparticles were studied assuming constant values of these coefficients.

## 2. Theoretical Model

We consider that the particle sizes satisfy the following condition: *a*_0_ << λ, where *a*_0_ is the linear size; and λ is the wavelength of light. Thus, we do not consider diffraction and light scattering processes. We also exclude the processes associated with particle sedimentation.

Let us consider a liquid-phase medium with nanoparticles irradiated by a light beam of intensity *I*_0_ that is uniformly distributed over a region (Figure 1).

Temperature and concentration gradients arise as a result of the action of the light field in the medium, and are then used to determine the heat- and mass-transfer processes (Soret effect). These phenomena are described by a system of balanced equations for the temperature and particles [26,27].

We define the system of balanced equations for heat conduction and the mass of nanoparticles transferred as follows:(1)$$C_{p}\rho \frac{\partial T}{\partial t}=\left(\lambda \left(C\right)\vec{gradT}\right)+\alpha \left(C\right)I_{0},$$
(2)$$\frac{\partial C}{\partial t}=\left(D\vec{gradC}\right)+D_{T}÷\left(C\left(1-C\right)\vec{gradT}\right)-\vec{V\cdot }\vec{gradC},$$
where *T* is the temperature of the medium; *C* is the volume concentration of the medium; *λ*(*C*) is the thermal conductivity of the medium; *α*(*C*) is the absorption coefficient of the light wave; *D* is the diffusion coefficient of nanoparticles; *D_T_* Tis the thermal diffusion coefficient; *V* is the concentration convection velocity; and *C_p_* and *ρ* are known thermophysical constants. It should be noted that, in Equation (2), we take into account the incompressibility of the nanofluid: $\vec{V}$ = 0 [27].

We now consider the one-dimensional case, neglecting the Dufour effect owing to its small contribution. We do not consider flows caused by the forces of pressure on the particles from the side of the light field. In further calculations, we assume:(3)$$\left(\lambda \left(C\right)\frac{\partial T}{\partial x}\right)\approx \lambda \left(C\right)\frac{\partial^{2}T}{\partial x^{2}},÷\left(D\frac{\partial C}{\partial x}\right)\approx D\frac{\partial C}{\partial x},$$
(4)$$\left(C\left(1-C\right)gradT\right)\approx C\left(1-C\right)\frac{\partial^{2}T}{\partial x^{2}}.$$

The validity of these approximations can be verified by direct calculations. We study the dynamics of nanoparticles against the background of the stationary temperature of the medium, i.e., ∂T/∂t = 0 (thermal processes are assumed to be 2–3 orders of magnitude faster than diffusion). We focus on processes with C≪1; this inequality ensures that the coagulation (coalescence) of nanoparticles can be disregarded.

According to theoretical and experimental studies [28,29], the concentration dependence of the thermal conductivity of a medium at low concentrations can be considered to be linear, as follows:(5)$$\lambda \left(C\right)=\lambda_{0}\left(1+pC\right),$$
where $\lambda_{0}$ is the value of the thermal conductivity coefficient of the fluid (without nanoparticles), and p is a linear coefficient. We consider the concentration dependence of the light absorption coefficient to be of the form: α = βC (where β exceeds zero). Given the stationary temperature regime, the approximations (Equations (4) and (5)), and low concentration, we obtain the following from the heat equation:(6)$$\frac{\partial^{2}T}{\partial x^{2}}=\frac{-\beta C}{\lambda_{0}\left(1+pC\right)}I_{0}\approx -\frac{\beta C}{\lambda_{0}}I_{0}\left(1-pC\right)\left(pC<1\right),$$

Using the approximations in Equations (3), (4) and (6), Equation (2) can be rewritten as follows:(7)$$\frac{\partial C}{\partial t}=D\frac{\partial^{2}C}{\partial x^{2}}-\frac{D_{T}\beta I_{0}}{\lambda_{0}}\left(1-pC\right)C^{2}-V\frac{\partial C}{\partial x},$$

For a complete description of the transport processes in the system under consideration, Equation (7) must be supplemented by the Navier-Stokes equation (to determine the velocity, *V*). In this case, the formulated problem can be solved numerically [25]. However, here, we use a different approach to derive the analytical solution. In particular, we represent the convective velocity in the following form:(8)$$V\left(C\right)=\frac{\eta \left(C\right)}{\rho \left(C\right)l},$$
where η(C) is the dynamic viscosity coefficient of the nanofluid; ρ(C) is its density; and *l* is the characteristic length of the system, the value of which is determined later.

We consider the dependence of the viscosity coefficient on concentration to be linear, such that:(9)$$\eta \left(C\right)=\eta_{0}\left(1+\gamma C\right),$$
where $\eta_{0}$ is the value of the viscosity coefficient of the base fluid devoid of nanoparticles. A similar dependence was obtained theoretically and experimentally, as confirmed in previous studies [28,29,30]. As for ρ(C), a linear dependence on concentration is also permissible here [31,32]:(10)$$\rho =\rho_{0}\left(1+\chi C\right),$$
where $\rho_{0}$ is the average density of the medium, and χ is the coefficient of the concentration expansion. As γ≫χ is real, we consider the density dependence on concentration to be insignificant.

Therefore, the expression for the velocity (Equation (8)) can be represented using Equations (9) and (10):(11)$$V\left(C\right)=\frac{\eta_{0}\left(1+\gamma C\right)}{\rho_{0}l\left(1+\chi C\right)}\approx \frac{\eta_{0}}{\rho_{0}l}\left(1+\gamma C\right),$$

As a result, the diffusion equation (Equation (7)) can be rewritten as follows:(12)$$\frac{\partial C}{\partial t}=D\frac{\partial^{2}C}{\partial x^{2}}-\frac{\eta_{0}}{l\rho_{0}}\left(1+\gamma C\right)\frac{\partial C}{\partial x}-\frac{D_{T}\beta I_{0}}{\lambda_{0}}C^{2}\left(1-pC\right),$$

We now introduce the dimensionless variables and parameterize Equation (12). As a result, we obtain:(13a)$$\frac{\partial C}{\partial \tau }=\frac{\partial^{2}C}{\partial y^{2}}-\delta \frac{\partial C}{\partial y}-\delta \gamma C\frac{\partial C}{\partial y}-C^{2}\left(1-pC\right),$$

The following notation is accepted here:(13b)$$\tau =\frac{S_{T}D\beta I_{0}}{\lambda_{0}}t,y=\frac{1}{\sqrt{b}}x,b=\frac{\lambda_{0}}{S_{T}\beta I_{0}},\sqrt{b}=l,\delta =\frac{\eta_{0}}{\rho_{0}D},$$

Thus, we demonstrate that light-induced thermal diffusion in nanofluids, in the low-particle-concentration approximation, against the background of a steady temperature, and taking into account concentration convection, can be described by nonlinear Equation (13a), which differs from the Burgers–Huxley equation [33] owing to the derivative in the last linear term.

First, we consider the two spatially homogeneous stationary states derived from $C^{2}\left(1-pC\right)=0$, which correspond to the roots of the equation, namely, $C_{1}=C_{2}=0,C_{3}=1/p\left(p>1\right)$. The kinetics of a dissipative system strongly depend on the stabilities of the stationary states. In our case, the states $C=C_{1,2}$ are twofold degenerate and unstable (they contain derivatives from the source $F^{\prime }\left(C\right)>0$), whereas state $C=C_{13}$ is stable. Thus, the medium studied herein is not bistable, unlike that studied by Ognev et al. [34].

We note that similar parabolic equations with cubic nonlinearities have been considered in previously published studies, in which they were applied to a model dissipative medium with arbitrary parameters [34], and to a nanofluid + radiation system [35,36]. We look for particular solutions in the form of the Cole-Hopf transform [36]:(14)$$C\left(y,\tau \right)=\frac{W_{y}^{\prime }}{W}\cdot \mu ,W=W\left(y,\tau \right),$$
where μ is a parameter, and ′ denotes the derivative.

By substituting Equation (14) into (13a) and equating the coefficients for the various powers of W to zero, we obtain an overdetermined system of equations for function W(y,μ):(15)$$W_{y\tau }^{\prime \prime }=W_{yyy}^{\prime \prime \prime }-\delta W_{yy}^{\prime \prime },$$
(16)$$W_{\tau }^{\prime }=3W_{yy}^{\prime \prime }+\delta \gamma \mu W_{yy}^{\prime \prime }+\left(\mu -\delta \right)W_{y}^{\prime },$$
(17)$$p\mu^{2}+\delta \mu +2=0,$$

From the last equation of this system, we obtain the values of parameter μ:(18)$$\mu_{1,2}=\frac{1}{2p},$$

The estimates of parameters γ and δ, which are provided below, show that roots $\mu_{1,2}$ are real. Furthermore, by integrating Equation (15) with respect to variable y, we obtain:(19)$$W_{\tau }^{\prime }=W_{yy}^{\prime \prime }-\delta W_{y}^{\prime }+C_{1}\left(\tau \right),$$

Using Equation (16), we obtain:(20)$$\left(2+\delta \gamma \right)W_{yy}^{\prime \prime }+\mu W_{y}^{\prime }+C_{1}\left(\tau \right)=0,$$

The solution for this equation can be represented as:(21)$$W\left(y,\tau \right)=\frac{C_{1}\left(\tau \right)}{\mu }y+C_{2}\left(\tau \right)+C_{3}\left(\tau \right)exp\left(-\omega y\right),$$
where ω=μ/(2+δγμ).

$C_{i}\left(\tau \right)$ can be determined using Equations (20) and (21), and it can be used to express the solution for function W(y,τ) as follows:(22)$$W\left(y,\tau \right)=\tilde{C_{1}}\left(\left(1-\frac{\delta }{\mu }\right)+\frac{1}{\mu }y\right)\tau +\tilde{C_{2}}+\tilde{C_{3}}exp\left(\omega \left(\omega +\delta \right)\tau -\omega y\right),$$
where $\tilde{C_{i}}$ are constants.

According to Equations (14) and (22), the desired concentration can be represented as:(23)$$C\left(y,\tau \right)=\mu \frac{\tilde{C_{1}}-\omega \tilde{C_{3}}exp\left(\omega \left(\omega +\delta \right)\tau -\omega y\right)}{{\overset{´}{C}}_{1}\left(\left(\mu -\delta \right)\tau +y\right)+\tilde{C_{2}}+\tilde{C_{3}}exp\left(\omega \left(\omega +\delta \right)\tau -\omega y\right)},$$

## 3. Solution Analysis

In this section, we examine the dependence of the determined exact solution (23) on parameters μ, ω, and δ. Parameter δ is estimated to be ≈ 10^5^ by assuming that $\eta_{0}$ = 10^−3^ kg/(m∙s), $\rho_{0}$= 10^3^ kg/m^3^, and *D* = 10^−11^ m^2^/s in Equation (13a). Furthermore, from the findings of previous studies [24,25,26], it follows that p ≈ 1 ÷ 1.5. Then, we must consider that $4p\ll \delta^{2}\gamma^{2}$ in Equation (18). Therefore, for sufficiently accurate roots $\mu_{1,2}$, we obtain $\mu_{1}=-2/\delta ,\omega_{1}=-2$, $\mu_{2}=-\delta /p,\omega_{2}=-1/p.$

Furthermore, by substituting the corresponding expressions for parameters and δ (their estimates) into Equation (23), we obtain two solutions:(24)$$C_{1}\left(y,\tau ,\mu_{1}\right)=\frac{2}{\delta \gamma }\frac{1+2c_{3}exp\left(-2\left(\delta \tau -y\right)\right)}{\left(\delta \tau -y\right)+c_{2}-c_{3}exp},$$
(25)$$C_{2}\left(y,\tau ,\mu_{2}\right)=\delta \gamma ,$$

Here, $c_{2}$ and $c_{3}$ are the newly redefined constants.

Let us consider Equation (24), the graph of which is shown in Figure 2; we see that it is a function of the variable traveling wave, z=(δτ−y). On the graph, the solution is presented in the form of soliton-like pulses moving to the right (with increasing time).

A distinctive feature of Equation (25) is the presence in the denominator of the (δ(γ/p−1)τ−y) term. Clearly, the nature of curve C(y) in Equation (25) strongly depends on the γ/p ratio. When plotting the function, we set γ/p = 3 (see Figure 3). Note that the wave-pulse profiles are not similar in this case.

The velocity of the wave front of Equation (24) can be determined using Equation (13b). As a result, we obtain:(26)$$\nu =\frac{\eta_{0}}{\rho_{0}\sqrt{b}}.$$

The speed depends on the thermodynamic, hydrodynamic, and optical characteristics of the nanofluid + radiation system. It follows from Equations (26) and (13b) that, in the case of anomalous thermal diffusion (S_T_ < 0—nanoparticles move in a higher-temperature region), the velocity acquires an imaginary term, which has no physical meaning. We believe that this case requires a separate consideration, which we plan to carry out in the future.

We numerically estimate the wavefront propagation velocity according to Equation (26) and consider water/silver as a nanofluid. Because the absorption coefficient β is present in Equation (26) through parameter b (see Equation (13b)), its evaluation requires the following equation [37]:$$\beta =\frac{12\pi }{\lambda }CIm\left(\frac{m^{2}-1}{m^{2}+2}\right),$$
where $m=m_{particles}/m_{fluid},m=n+ik.$

Here, by assuming $m_{p}=0.15+3.5i,\;m_{f}=1.33+0.2i$, $\lambda =6.5\cdot 10^{-7}$ m, and $C=1\cdot 10^{-4}$, we obtain $\beta \approx 2\cdot 10^{3}$. Furthermore, by substituting $\eta_{0}=0.003$ kg/m⋅s, $\rho_{0}=1\cdot 10^{3}$, $I_{0}=1\cdot 10^{5}$ W/m^2^, and $\lambda_{0}$ = 0.5 W/m⋅K into Equations (26) and (13b), we obtain a velocity estimate: $v\approx 2\cdot 10^{-3}$ m/s. The estimated velocity value depends on the initial concentration distribution, which can be obtained from Equation (24) for τ = 0. It should be noted that our approach cannot solve the problem analytically under arbitrary initial conditions.

## 4. Conclusions

Two exact analytical solutions of a nonlinear one-dimensional Burgers–Huxley-type equation were obtained. These solutions describe the dynamics of the concentration of nanoparticles in a liquid-phase medium by taking into account concentration convection. In this case, the coefficients of thermal conductivity, viscosity, and absorption of radiation by particles were found to be concentration dependent.One of the solutions found was represented as a solitary wave. Both solutions were expressed in the form of traveling single-phase waves.In the framework of the formulated approximations, the nanofluid + radiation system under study exhibited one stable (doubly degenerate) state and one unstable state. It should be noted that convection does not affect the nature of the stability.Within the framework of these approximations, it is possible to obtain the spatiotemporal dependence of the particle absorption coefficient, which exhibits the same wave characteristics as those of Equations (24) and (25).

Invariably, we did not consider some issues. In particular, based on the fact that the equation under consideration is autonomous, studying the equation on the phase plane is of significant interest and will be the subject of our further research.

## Figures and Tables

**Figure 1 nanomaterials-11-01291-f001:**
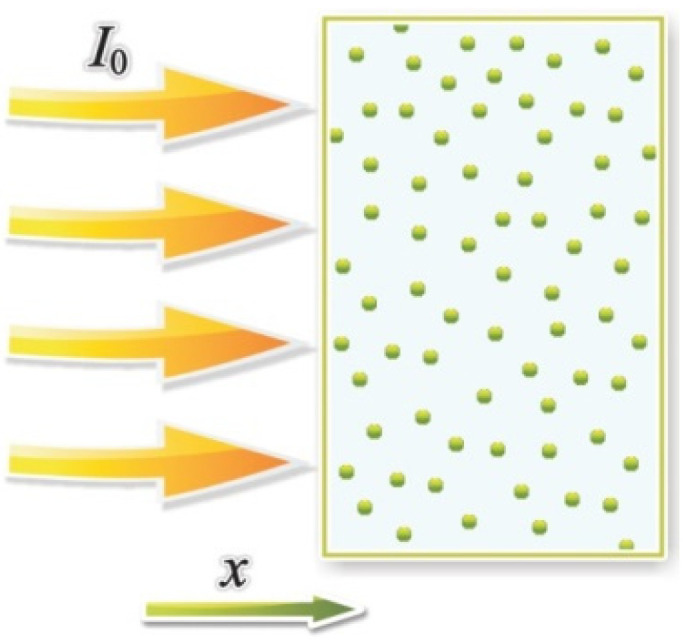
Geometry of the problem.

**Figure 2 nanomaterials-11-01291-f002:**
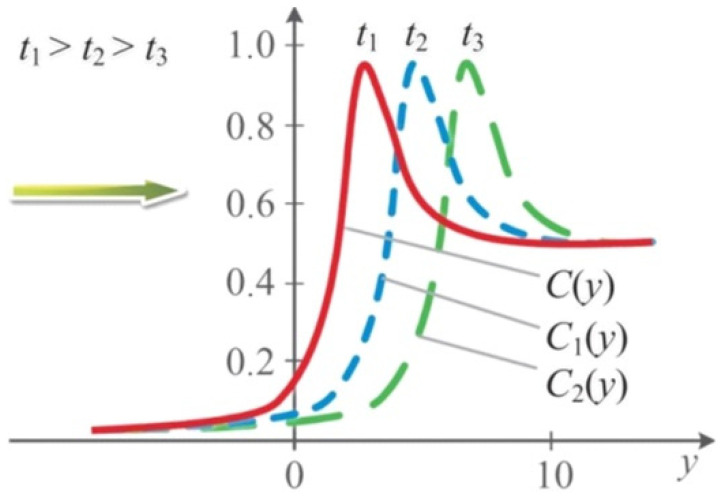
Solution of Equation (24).

**Figure 3 nanomaterials-11-01291-f003:**
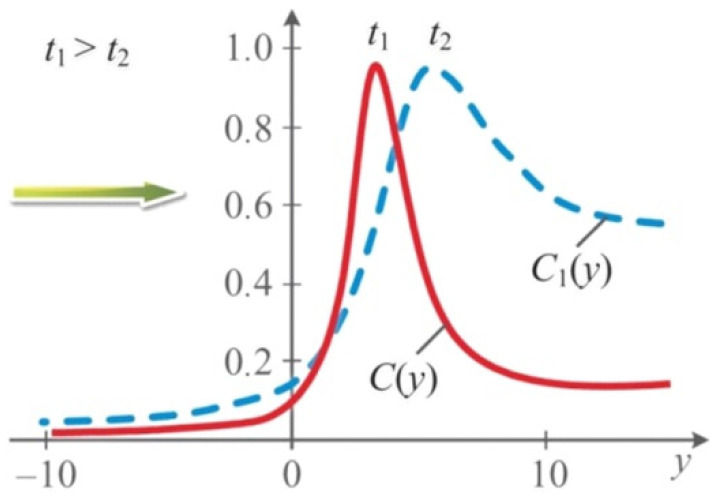
Solution profiles of the concentration wave obtained from Equation (25) with increasing time.

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
