# Peer review of "Appearance of a Solitary Wave Particle Concentration in Nanofluids under a Light Field"

_nanomaterials, 2021, doi:10.3390/nano11051291_

Round 1

Reviewer 1 Report

The contents are relevant to this journal. The main observations are listed below. I can recommend its publication but before publication, I suggest following revision. 

  1. Authors should not be overused the well-known information in the abstract, even as background information. The abstract should be briefly written to describe the purpose of the research, the principal results, and major conclusions. Authors should revise it.
  2. In the Introduction, the literature review was not logically organized and all literatures cited seem separate descriptions without connections. The readers can’t know what the state-of-art methodologies or algorithms in two-phase flow study are, what difficulties in two-phase flow study are, what problems or gaps the current study plans to resolve or fill, and how significant or what contribution the current study is?
  3. The authors should do a better job on commenting the results. A reasonable physical explanation should be provided for the observed trends, not only report what is graphically seen in the figures. More physical insight of the Discussion section is needed.
  4. Many mathematical formulas are expressed. The author should summarize them.
  5. Reference list is not uniform.  Author should use ISO abbreviation for Journals names. Also provide the appropriate reference of governing equations.
  6. There are few language and typographic errors, for instance Result and discussions should be Results and discussion. Typographic errors and word spacing can be fixed in the revised version for running a spell check.
  7. Make the font and font size consistent in the entire manuscript.
  8. I would also suggest that author should add few most relevant papers published in this journal during the last 2 years

Author Response

Dear reviewer,

thank you very much for your comments.

Below you may find our answers to your comments and suggestions.

  1. Authors should not be overused the well-known information in the abstract, even as background information. The abstract should be briefly written to describe the purpose of the research, the principal results, and major conclusions. Authors should revise it.

Response: Following your suggestion, we removed the first sentence of the abstract and added some relevant content (lines 17-20, 25-26). We believe that all other necessary components of the abstract are already included.

  1. In the Introduction, the literature review was not logically organized and all literatures cited seem separate descriptions without connections. The readers can’t know what the state-of-art methodologies or algorithms in two-phase flow study are, what difficulties in two-phase flow study are, what problems or gaps the current study plans to resolve or fill, and how significant or what contribution the current study is?

Response: We added relevant references [13-18] and described in more detail the state-of-art methodologies and algorithms proposed for the study of the two-phase flow (lines 36-51, 64-66).

  1. The authors should do a better job on commenting the results. A reasonable physical explanation should be provided for the observed trends, not only report what is graphically seen in the figures. More physical insight of the Discussion section is needed.

Response: In the revised manuscript, we removed the following paragraph: Turning to the explicit expression for parameter b (see equation (13)), we particularly note that the convective velocity is proportional to both the light intensity and radiation absorption coefficient».

Further, we inserted the following paragraph:

“It follows from equations (26) and (13a) that in the case of anomalous thermal diffusion (ST <0 – nanoparticles move in a region with a higher temperature), the expression for the velocity becomes imaginary, which has no physical meaning. We believe that this case requires a separate consideration, which we plan to carry out in the future.” (lines 182-186, 196-198).

  1. Many mathematical formulas are expressed. The author should summarize them.

Response: The model description of the nanocolloid behavior involves the use of mass and heat transfer equations. In this paper, we have tried to include detailed calculations so that readers can easily understand the thread of reasoning.

  1. Reference list is not uniform.  Author should use ISO abbreviation for Journals names. Also provide the appropriate reference of governing equations.

Response: Thank you for pointing this out. The reference list has been appropriately formatted by OSA Language Editing.

  1. There are few language and typographic errors, for instance Result and discussions should be Results and discussion. Typographic errors and word spacing can be fixed in the revised version for running a spell check.

Response: Thank you for pointing this out. We had the paper proofread and revised by OSA Language Editing.

  1. Make the font and font size consistent in the entire manuscript.

Response: We asked OSA Language Editing to ensure that the font type and size are consistent throughout the manuscript.

  1. I would also suggest that author should add few most relevant papers published in this journal during the last 2 years.

Response: We added two more recent references published in Nanomaterials (Ref. 16 and 18).

Reviewer 2 Report

The introduction needs to be expanded. The authors should discuss relevant previous work to illustrate the motivation and scientific advancement of the current work. Suggested articles to discuss: https://doi.org/10.1016/j.physleta.2020.126500; https://doi.org/10.1007/s10973-020-09590-2; https://doi.org/10.1016/j.jclepro.2019.119378; https://doi.org/10.1016/j.renene.2020.08.039

Rather than just presenting the model and the analysis, the authors need to expand significantly the discussion part of the manuscript. What are the implications of this study? How can we benefit from the model and the solution and apply to future design? What are the limitations of the model application?

The authors must address these issues before publication. 

Author Response

Dear reviewer,

thank you very much for your comments.

Below you may find our answers to your comments and suggestions.

  1. The introduction needs to be expanded. The authors should discuss relevant previous work to illustrate the motivation and scientific advancement of the current work. Suggested articles to discuss: https://doi.org/10.1016/j.physleta.2020.126500; https://doi.org/10.1007/s10973-020-09590-2; https://doi.org/10.1016/j.jclepro.2019.119378; https://doi.org/10.1016/j.renene.2020.08.039

Response: We added suggested references (13-18) and expanded the introduction (lines 38-53). 

  1. Rather than just presenting the model and the analysis, the authors need to expand significantly the discussion part of the manuscript. What are the implications of this study? How can we benefit from the model and the solution and apply to future design? What are the limitations of the model application?

The authors must address these issues before publication. 

Response: We expanded the sections Solution analysis at the end of the paragraph (lines 185-189, 199-202).

Round 2

Reviewer 1 Report

The paper in my opinion has been widely improved. Both the theoretical aspects and the application leads to suitable results.

Reviewer 2 Report

The revision looks good and can be accepted for publication.